# Tailed giant Tupanvirus possesses the most complete translational apparatus of the known virosphere

Jônatas Abrahão[1,2], Lorena Silva[1,2], Ludmila Santos Silva[1,2], Jacques Yaacoub Bou Khalil[3], Rodrigo Rodrigues[2], Thalita Arantes[2], Felipe Assis[2], Paulo Boratto[2], Miguel Andrade[4], Erna Geessien Kroon[2], Bergmann Ribeiro [4], Ivan Bergier [5], Herve Seligmann[1], Eric Ghigo[1], Philippe Colson[1], Anthony Levasseur[1], Guido Kroemer[6,7,8,9,10,11,12], Didier Raoult[1] & Bernard La Scola[1]

Here we report the discovery of two Tupanvirus strains, the longest tailed *Mimiviridae* members isolated in amoebae. Their genomes are 1.44–1.51 Mb linear double-strand DNA coding for 1276–1425 predicted proteins. Tupanviruses share the same ancestors with mimivirus lineages and these giant viruses present the largest translational apparatus within the known virosphere, with up to 70 tRNA, 20 aaRS, 11 factors for all translation steps, and factors related to tRNA/mRNA maturation and ribosome protein modification. Moreover, two sequences with significant similarity to intronic regions of 18 S rRNA genes are encoded by the tupanviruses and highly expressed. In this translation-associated gene set, only the ribosome is lacking. At high multiplicity of infections, tupanvirus is also cytotoxic and causes a severe shutdown of ribosomal RNA and a progressive degradation of the nucleus in host and non-host cells. The analysis of tupanviruses constitutes a new step toward understanding the evolution of giant viruses.

[1] MEPHI, APHM, IRD 198, Aix Marseille Univ, IHU-Méditerranee Infection, 19-21 Bd Jean Moulin, 13005 Marseille, France. [2] Laboratório de Vírus, Instituto de Ciências Biológicas, Departamento de Microbiologia, Universidade Federal de Minas Gerais, Belo Horizonte 31270-901, Brazil. [3] CNRS, 13005 Marseille, France. [4] Laboratório de Microscopia Eletrônica e Virologia, Departamento de Biologia Celular, Instituto de Ciências Biológicas, Universidade de Brasília, Asa Norte, Brasília 70910-900, Brazil. [5] Lab. Biomass Conversion, Embrapa Pantanal, R. 21 de Setembro 1880, 79320-900 Corumbá/MS, Brazil. [6] Cell Biology and Metabolomics Platforms, Gustave Roussy Cancer Campus, Villejuif 94805, France. [7] Equipe 11 labellisée Ligue Nationale contre le Cancer, Centre de Recherche des Cordeliers, Paris 75006, France. [8] Institut National de la Santé et de la Recherche Médicale (INSERM), Paris 75654, France. [9] Université Paris Descartes, Sorbonne Paris Cité, Paris 75015, France. [10] Université Pierre et Marie Curie, Paris 75005, France. [11] Pôle de Biologie, Hôpital Européen Georges Pompidou, AP-HP, Paris 75015, France. [12] Department of Women's and Children's Health, Karolinska University Hospital, Stockholm SE-171 76, Sweden. Jônatas Abrahão, Lorena Silva and Ludmila Santos Silva contributed equally to this work. Correspondence and requests for materials should be addressed to D.R. (email: didier.raoult@gmail.com) or to B.S. (email: bernard.la-scola@univ-amu.fr)

Translation is one of the canonical frontiers between the cell world and the virosphere. Even the simplest non-viral obligatory intracellular parasites present a wealthy set of translation apparatus, including aminoacyl tRNA synthetases, tRNAs, peptide synthesis factors and ribosomal proteins[1–3]. The nature of the parasitism of the most of non-viral obligatory intracellular parasites relies on partial or severe deficiency of genes related to energy production. Although sharing a similar lifestyle with those organisms, the most of the virus lack not only energy production genes, but also translation-related genes[1–3].

In this context, the discovery of mimiviruses and other amoeba-infecting giant viruses surprised the scientific community due their unusual sizes and long genomes, able to encode from hundreds to thousands of genes, including tRNAs, peptide synthesis factors and, for the first time seen in the virosphere, aminoacyl tRNA synthetases (aaRS)[4]. Although the very first discovered mimivirus (*Acanthamoeba polyphaga mimivirus*) encodes four different types of aaRS (Arg, Cys, Met, TyrRS), other mimivirus isolates genomes were described, containing up to seven types of aaRS, as Megavirus chilensis and LBA111 (Arg, Asn, Cys, Ile, Met, Trp, TyrRS)[5]. While amoeba-infecting mimiviruses present up to six copies of tRNA, *Cafeteria roenbergensis* virus (CroV), a group II algae-infecting of *Mimiviridae*, encodes 22 sequences for five different tRNAs[5]. Even more surprisingly, metagenomics data reveal that klosneuvirus genome may encodes aaRS with specificities for 19 different amino acids, over 10 translation factors and several tRNA-modifying enzymes[6]. However, klosneuviruses were not isolated and therefore there is no data regarding important biological features as virus particles, morphogenesis and host-range.

Here we report the discovery of two Tupanvirus strains, the longest tailed *Mimiviridae* members isolated in amoebae. Their genomes are 1.44–1.51 Mb linear double-strand DNA coding for 1276–1425 predicted proteins. Tupanviruses share the same ancestors with mimivirus lineages and these giant viruses present the largest translational apparatus within the known virosphere, with up to 70 tRNA, 20 aaRS, 11 factors for all translation steps, and factors related to tRNA/mRNA maturation and ribosome protein modification. Moreover, two sequences with significant similarity to intronic regions of 18 S rRNA genes are encoded by the tupanviruses and highly expressed. In this translation-associated gene set, only the ribosome is lacking. Tupanvirus is also cytotoxic and causes a severe shutdown of ribosomal RNA and a progressive degradation of the nucleus in host and non-host cells. The analysis of tupanviruses constitutes a new step towards understanding the evolution of giant viruses.

## Results

**Tupanviruses description and cycle.** While attempting to find new and distant relatives of currently known giant viruses, we performed prospecting studies in special environments. Soda lakes (Nhecolândia, Pantanal biome, Brazil) are known as environments that conserve and/or mimic ancient life conditions (extremely high salinity and pH) and are considered some of the most extreme aquatic environments on Earth[7]. We also prospected giant viruses in ocean sediments collected at a depth of 3000 m (Campos dos Goytacazes, Brazilian Atlantic Ocean).

Both in soda lake and deep ocean samples, we found optically visible *Mimiviridae* members that surprisingly harbored a long, thick tail (Fig. 1a, b) as they grew on *Acanthamoeba castellanii* and *Vermamoeba vermiformis*. We named these strains Tupanvirus soda lake and Tupanvirus deep ocean as a tribute to the South American Guarani Indigenous tribes, for whom Tupan—or Tupã—(the God of Thunder) is one of the main mythological figures. Electron microscopy analyses revealed a remarkable virion structure. Tupanviruses present a capsid similar to that of amoebal mimiviruses in size (~450 nm) and structure, including a stargate vertex and fibrils[8] (Figs. 1a–d, 2a–q). The Tupanvirus virion presents a large cylindrical tail (~550 nm extension; ~450 nm diameter, including fibrils) attached to the base of the capsid (Figs. 1b–d, 2a,i–k). This tail is the longest described in the virosphere[9]. Microscopic analysis suggests that the capsid and tail are not tightly attached (Figs. 1e, f, 2a, i; Supplementary Movies 1 and 2), although sonication and enzymatic treatment of purified particles were not able to separate the two parts (Fig. 2f–h). The average length of a complete virion is ~1.2 μm, although some particles can reach lengths of up to 2.3 μm because of the variation in the tail's size; this makes them the one of the longest viral particles described to date (Figs. 1, 2i–k). Furthermore, there is a lipid membrane inside the capsid, which is associated with the fusion with the cellular membrane and the release of capsid content (Fig. 1f). Tail content appears to be released after an invagination of the phagosome membrane inside the tail (Fig. 1e). In contrast

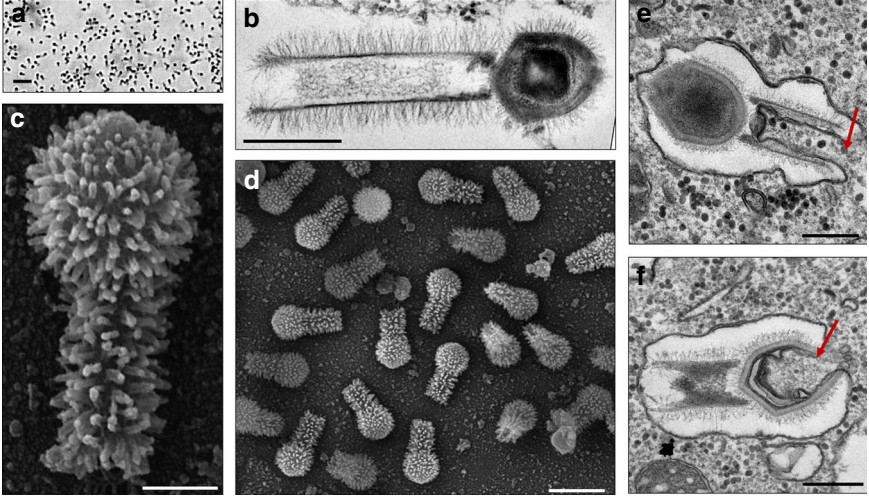

**Fig. 1** Tupanvirus soda lake particles and cycle. **a** Optical microscopy of Tupanvirus particles after haemacolour staining (1000 × ). Scale bar, 2 μm. **b** Super particle (>1000 nm) observed by transmission electron microscopy (TEM). Scale bar, 500 nm. **c, d** Scanning electron microscopy (SEM) of Tupanvirus particles. Scale bars 250 nm and 1 μm, respectively. **e, f** The initial steps of infection in *A. castellanii* involve the release of both capsid (**e**) and tail (**f**) content into the amoeba cytoplasm (red arrows). Scale bars, 350 nm and 450 nm, respectively

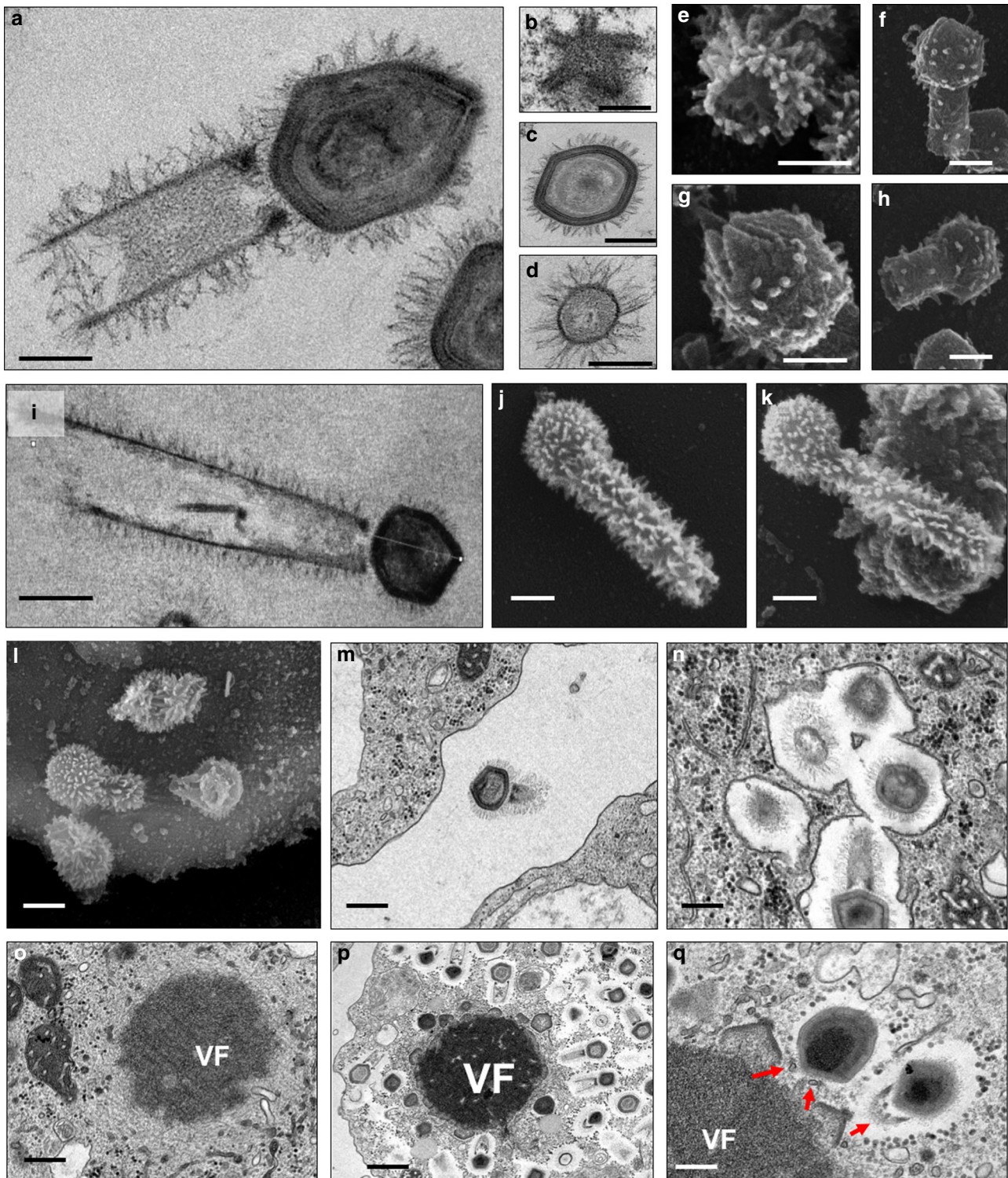

**Fig. 2** Tupanvirus soda lake particles and cycle features. **a** Transmission electron microscopy (TEM) highlights the inner elements of the whole particle. Scale bar, 200 nm. **b** Star-gate vertex transversally cut. Scale bar, 100 nm. **c** Capsid transversally cut. Scale bar, 100 nm. **d** Tail transversally cut. Scale bar, 200 nm. **e–h** Scanning electron microscopy (SEM) of purified particles. Scale bars, 250 nm. The treatment of particles with lysozyme, bromelain and proteinase-K removed most of the fibers, revealing head and tail junction details. Super particles (>2000 nm) could be observed by TEM (**i**) and SEM (**j**, **k**). Scale bars, 400 nm. Cycle steps are shown from **l–r**. **l** Viral particle attachment to *Acanthamoeba castellanii* surface; scale bar, 500 nm; **m** phagocytosis; scale bar, 500 nm; **n** particles in a phagosome; scale bar, 500 nm; **o** early viral factory; scale bar, 500 nm; and **p**, **q** mature viral factories. Scale bars 1 μm and 250 nm, respectively. Arrows highlight tail formation associated with the viral factories. VF viral factory

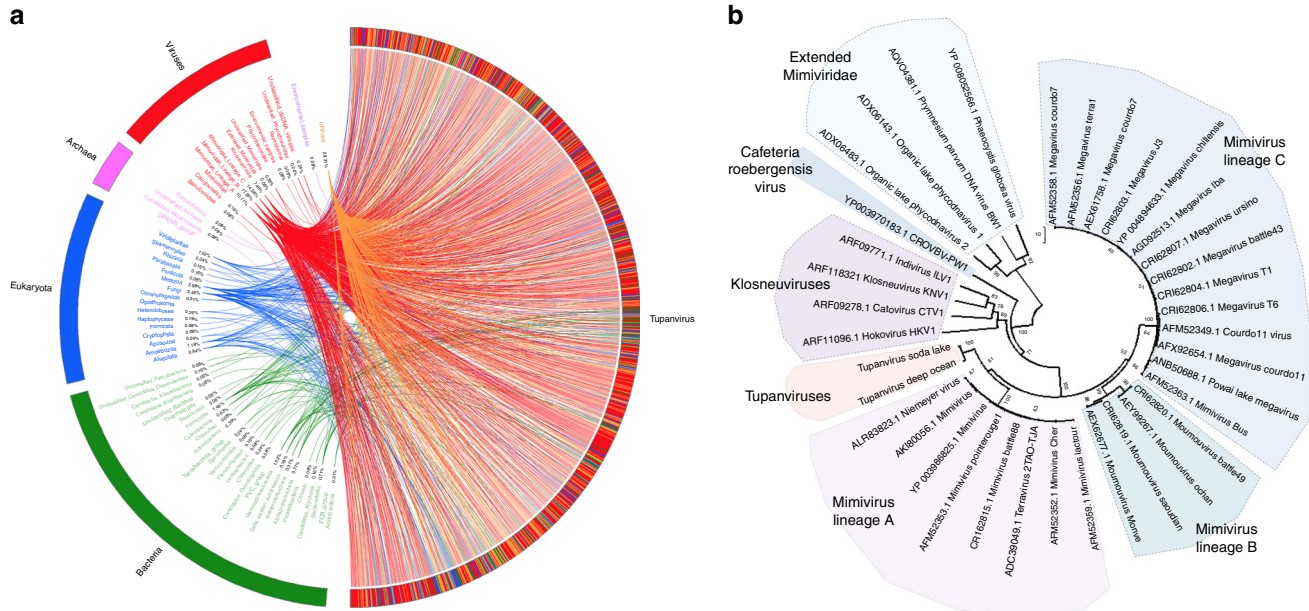

**Fig. 3** Tupanvirus soda lake rhizome and Mimiviridae family B DNA polymerase tree. **a** The rhizome shows that most Tupanvirus soda lake genes have mimiviruses as best hits. However, correspondence among Tupanvirus and *Archaea*, *Eukaryota*, *Bacteria* and other viruses was also observed. **b** Family B DNA polymerase maximum likelihood phylogenetic tree demonstrating the position of Tupanvirus among *Mimiviridae* members, likely forming a new genus

to other giant viruses[10–14], Tupanviruses similarly replicate both in *A. castellanii* and *V. vermiformis*. Viral particles attach to the host-cell surface and enter through phagocytosis (1 h.p.i.) (Fig. 2 l,m,n). The inner membrane of the capsid merges with the amoebal host phagosome membrane, releasing the genome (2–6 h.p.i.) (Fig. 1f). A viral factory (VF) is then formed (7–12 h. p.i.)[15] where particle morphogenesis occurs (Fig. 2o–q; Supplementary Movies 1 and 2). During this step, the virion tail is attached to the capsid after its formation and closure. Later in the process (16–24 h.p.i.), the amoebal cytoplasm is filled with viral particles, followed by cell lysis and the release of viruses (Supplementary Fig. 1). This nucleus-like viral factory has also recently been reported in bacteria and fuels the concept of a virocell[16,17]. In that perspective, viral factories actively producing the progeny could be considered as the nuclei of virocells[16,17].

**The genomes of tupanviruses.** The tupanvirus soda lake and tupanvirus deep ocean genomes (GenBank accession number KY523104 and MF405918) are linear dsDNA molecules measuring 1,439,508 bp and 1,516,267 bp (GC% ~28%), respectively—the fourth largest viral genomes described to date[10,18]—containing a total of 1276 and 1425 predicted ORFs, 375 and 378 of which are ORFans (ORFs with no matches in current databases), respectively. To date, the largest genomes belong to pandoravirus isolates, and the largest one, *P. salinus*, has 2,473,870 bp and encodes 2,556 putative proteins[10]. The rhizome[19] of tupanvirus (graphical representation of gene-by-gene best hits) revealed sequences from mimiviruses of amoebae (~42%) and klosneuviruses (~8%) as their main best hits. Other best hits were mostly sequences from eukaryotes (~11%) and bacteria (~8%) (Fig. 3a; Supplementary Fig. 2A). Tupanviruses exhibited relatively close numbers of best matches to amoebal mimiviruses from lineages A (~10%), B (~18%) and C (~14%). These data and phylogenetic analyses demonstrate that they cluster with amoebal mimiviruses, suggesting that tupanviruses are distant relatives of, and comprise a sister group to, mimiviruses of amoeba (Figs. 3b, 4a). The 'AAAATTGA' promoter motif was found ~410 times in Tupanvirus deep ocean intergenic regions, a frequency similar to that of

other *Mimiviridae* members, and ~600% more frequent than those coding regions ($p < 10^{-95}$, Fisher exact test)[20–22] (Fig. 4b). The pangenome of the family *Mimiviridae*, when taking into account the gene contents of tupanviruses, klosneuviruses and distant relatives to amoebal mimiviruses, was found by Proteinortho to comprise 8,753 groups of orthologues ($n = 3588$) or virus unique genes ($n = 5165$). A total of 189 groups were shared by at least one tupanvirus; one mimivirus of *Acanthamoeba* of each lineage A, B and C; and one klosneuvirus, and 100 of them were also shared with Cafeteria roenbergensis virus. In addition, a total of 757 groups were shared by a tupanvirus and at least another mimivirus: 477 were shared with Megavirus chiliensis, 434 with Moumouvirus, 431 with Mimivirus, 287 with Klosneuvirus, 126 with Cafeteria roenbergensis virus, and 59 with Phaeocystis globosa virus 12 T. Among these 757 groups, 583 corresponded to clusters of orthologous groups previously delineated for mimiviruses[19]. Finally, a total of 775 tupanvirus genes were absent from all other mimivirus genomes (Supplementary Data 1).

**Proteomic analysis.** Proteomic analysis of Tupanvirus soda lake particles revealed 127 proteins, nearly half (67/127 = 52.8%) of which are unknown and eight of which are encoded by ORFans (11/127 = 8.6%) (Supplementary Data 2; Supplementary Note 1). Although 62 Tupanvirus virion proteins homologous were not found by proteomics, either in Mimivirus or in *Cafeteria roenbergensis virus* particles, there are no distinct clues about which protein (s) could be associated with the tupanvirus fibrils and tail structure.

**The most complete translational apparatus of the virosphere.** Analyses of the tupanvirus gene sets related to energy production revealed a clear dependence of these viruses on host energy production mechanisms, similarly to other mimiviruses, because genes involved in glycolysis, the Krebs cycle and the respiratory chain are mostly lacking[22–24]. Astonishingly, Tupanvirus soda lake and Tupanvirus deep ocean exhibit the largest viral sets of genes involved in translation, with 20 ORFs related to aminoacylation (aaRS) and transport, and 67 and 70 tRNA associated with 46 and 47 codons, respectively (Fig. 5a; Supplementary

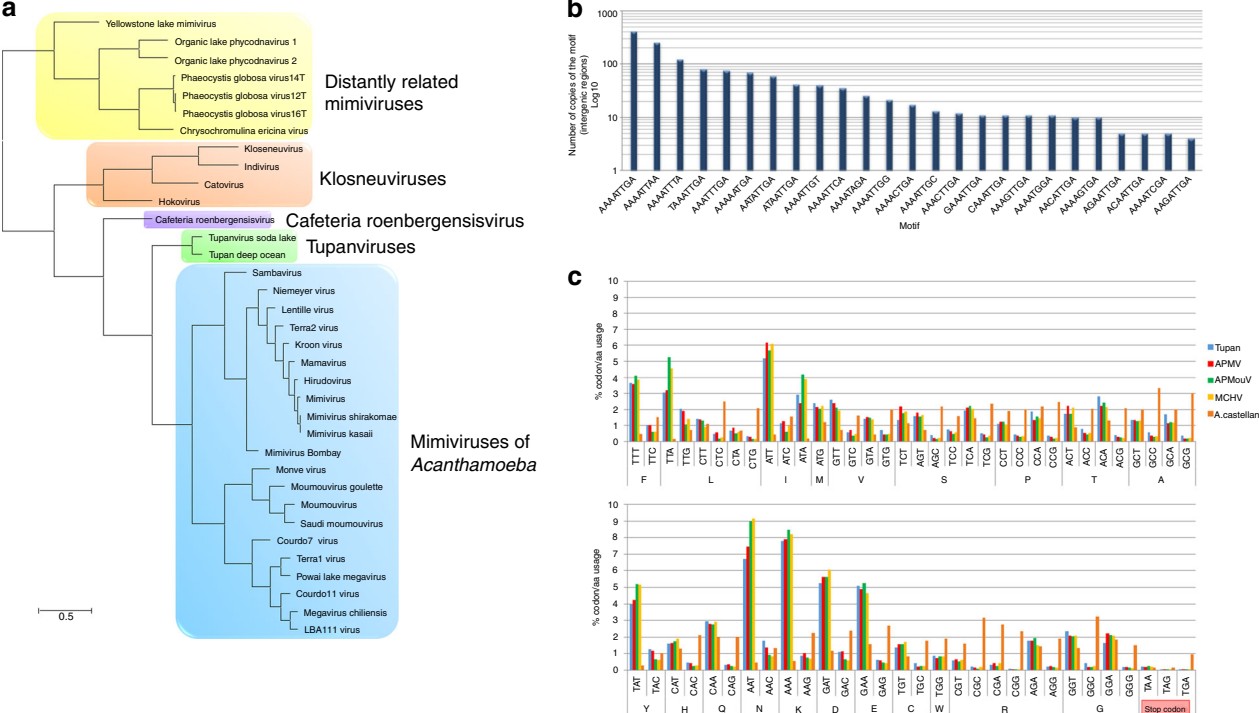

**Fig. 4** Tupanvirus soda lake hierarchical clustering, promoter's motifs and amino-acid usage analysis. **a** Hierarchical clustering tree based on presence-absence matrix of cluster of orthologous genes shared by *Mimiviridae* members. **b** Frequency of mimivirus AAAATTGA canonical promoter motifs in Tupanvirus intergenic regions. We also analyzed the presence of the AAAATTGA motif with SNPs, considering each motif position. **c** Comparative amino-acid usage analysis of Tupan, *A. castellanii* and mimivirus lineages **a**, **b** and **c**. The amino-acid usage for protein sequences was calculated using the CGUA (General Codon Usage Analysis) tool

Fig. 2B, C, 3 and 7; Supplementary Datas 3 and 4). Tupanvirus deep ocean encodes an ambiguous tRNA related to the rare amino-acid pyrrolysine. Only selenocysteine-related genes are lacking, as previously observed for many cellular organisms[25]. Several translation factors were identified, including eight translation initiation factors (IF2 alpha, IF2 beta, IF2 gamma, IF4e (2 copies in Tupanvirus soda lake), IF5a (2 copies in Tupanvirus deep ocean), SUI1, IF4a), one elongation/initiation factor (GTP-binding elongation/initiation), one elongation factor (Ef-aef-2), and one release factor (ERF1) (Fig. 5a; Supplementary Fig 2B, C and 3; Supplementary Datas 3 and 4). Furthermore, we detected additional translation-related genes: factors related to tRNA maturation and stabilization (tRNA nucleotidyltransferase, tRNA guanylyltransferase, cytidine deaminase, RNA methyl transferase); mRNA maturation (poly(A) polymerase, mRNA capping enzyme) and splicing (RNA 2—phosphotransferase Tpt1 family protein); and ribosomal protein modification (ribosomal-protein-alanine N-acetyltransferase, FtsJ-like methyl transferase) (Fig. 5a; Supplementary Fig 2B; Supplementary Datas 3, 4). Phylogenies based on aaRS are acknowledged to be complex as they depend on sequence sampling, and natural and canonical taxonomical groups can be separated into different clades, as previously observed[6]. To reduce such disturbances in the trees during their construction, we selected the 100 best hits related to Tupanvirus soda lake, plus the sequences of 5–10 amoebozoa and those of klosneuviruses. In most of the trees, sequences from natural clades were clustered together, although some amoebozoa were separated into different clusters. Based on the 20 aaRS trees, it is not possible to state that the origin of most of these tupanvirus aaRS genes is cellular (Supplementary Fig. 4). Furthermore, the mosaic structure of aaRS gene reinforced the difficult to state on the origin of these genes, as illustrated in Supplementary Fig. 5. This contrasting result to that reported by Schultz et al.[6], 2017 for

klosneuviruses may also be explained by the different sampling used for alignments and trees construction. In addition, the codon and amino-acid usage of tupanviruses is substantially different from that of *Acanthamoeba* spp. In tupanviruses, we observed a high correlation between tRNA isoacceptors and the most used codons, as these viruses present more tRNA isoacceptors for highly abundant codons (Fig. 4c; Supplementary Figs. 2C and 3). Surprisingly, we found two different copies of an 18 S rRNA intronic region in tupanviruses (Supplementary Fig. 6; Fig. 6a–e). In fact, such 18 S rRNA intronic regions are widespread in all mimiviruses (lineages A and B present only one copy in an intronic region and next to a self-splicing group I intron endonuclease) but are not found in klosneuviruses. Phylogenetic analyses revealed that the two copies found in tupanviruses had separate and different origins (Fig. 6f). Although Tupanvirus 18 S rRNA intronic sequences are located in intergenic regions, qPCR and FISH demonstrated that Tupanvirus soda lake 18 S rRNA intronic sequences are highly expressed during the entire replicative cycle but particularly during intermediate and late phases (6 and 12 h post-infection) (Fig. 7a; Supplementary Fig. 8). Furthermore, Tupanvirus soda lake is more tolerant to the translation-inhibiting drugs geneticin and cycloheximide than Mimivirus, an impressive characteristic considering the natural ribosomal RNA shutdown it performs in the permissive host (Fig. 8h). The functions of these 18 S rRNA intronic region sequences require further clarification. No exonic region of 18 S rRNA was found in the genomes of tupanviruses or any previously described mimivirus. The comparison between contents in translation-related categories of genes present in tupanviruses and cellular organisms reveals that tupanviruses present a richer gene set than *Candidatus Carsonella ruddii* (Bacteria) and *Nanoarchaeum equitans* (Archaea) (not considering ribosomal proteins). Tupanvirus deep ocean has even

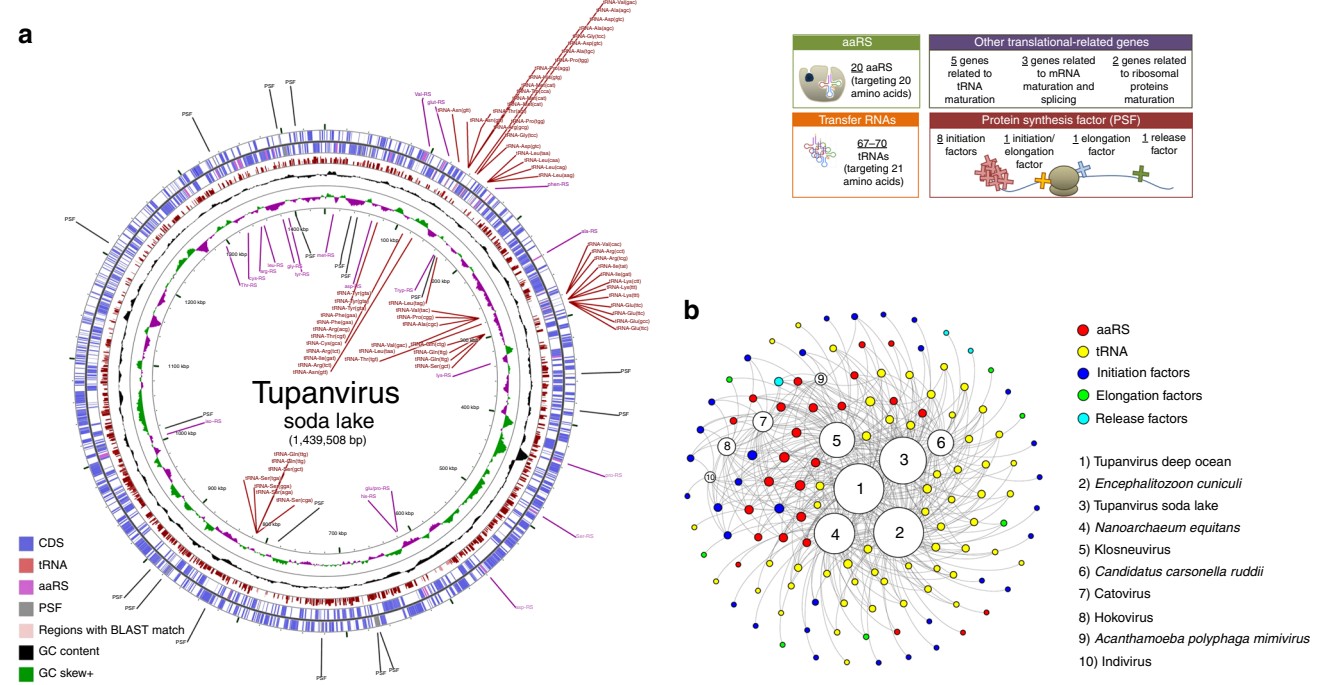

**Fig. 5** Tupanvirus genome-translation-related factors. **a** Circular representation of Tupanvirus soda lake genome highlighting its translation-related factors (aaRS, tRNAs and PSF). The box (upright) summarizes this information and considers the Tupanvirus deep ocean data set. **b** Network of shared categories of translation-related genes (not considering ribosomal proteins) present in tupanviruses, Mimivirus (APMV), Klosneuvirus, Catovirus, Hokovirus, Indivirus and cellular world organism—*Encephalitozoon cuniculi* (*Eukaryota*), *Nanoarchaeum equitans* (*Archaea*) and *Candidatus Carsonella ruddii* (*Bacteria*). The diameter of the organism's circles (numbers) is proportional to the number of translation-related genes present in those genomes. CDS coding sequences, tRNA transfer RNA, aaRS aminoacyl tRNA synthetase, PSF protein synthesis factors

more such genes than *Encephalitozoon cuniculi*, a eukaryotic organism (Fig. 5b). Even if the impressive translation gene set recently described for klosneuviruses is taken into account, tupanviruses are the first viruses reported to harbor the complete set of the 20 aaRS (Fig. 5; Supplementary Figs. 3, 6 and 7).

**Host range and host-ribosomal shutdown characterization.** In contrast to other mimiviruses, Tupanvirus soda lake was able to infect a broad range of protist organisms in vitro. Surprisingly, we observed four distinct profiles of infectiveness: (i) productive cycle in permissive cells; (ii) abortive cycle; (iii) refractory cells; and (iv) most surprisingly, non-host cells exhibiting a cytotoxic phenotype in the presence of Tupanvirus without multiplication, a circumstance never previously reported—to the best of our knowledge (Supplementary table 1; Supplementary Notes 1). This latter profile was intriguing because toxicity was observed in *Tetrahymena* sp., a ravenous free-living protist[26]. This unusual phenotype was also observed in *A. castellanii* but only at higher multiplicities of infection (50 and 100; Fig. 8a). No such cytotoxicity was observed for Mimivirus (Fig. 8a). This toxic profile associated specifically with Tupanvirus appears to be related to shutting down host ribosomal RNA abundance, insofar that Tupanvirus leads to a reduction of host rRNA amounts by a mechanism not related to the canonical ribophagy/autophagy process (Figs. 7b–d, 8b–d; Supplementary Fig. 9). A remarkable acidification of amoebal cytoplasm is induced by Tupanvirus infection (but not mimivirus) (Fig. 7b; Supplementary Fig. 9A, B). The treatment of acanthamoeba with chloroquine, a lysosomal toxin, or bafilomycin did not prevent the Tupanvirus-induced rRNA shutdown (Supplementary Fig. 9A, C–H). Transfection with an siRNA targeting Atg8-2, a protein required for ribophagy/autophagy, failed to prevent Tupanvirus-induced

ribosomal shutdown in *A. castellanii* (Fig. 7c, d). Transmission electron microscopy (TEM) of vesicles containing ribosomes after Tupanvirus infection (2 h.p.i.) revealed that these structures were formed close to the nuclear membrane after invagination and engulfment of ribosomes, mostly by single-membrane vesicles, rarely by double-membrane vesicles (Fig. 7e). These small vesicles then aggregated, accumulating more ribosomes and forming large structures containing ribosomes (Fig. 8d), which were fully depleted from the amoebal cytoplasm 6–9 h.p.i, when strong cellular rRNA shutdown could be detected (Fig. 8c). In addition, Tupanvirus infection induces nuclear/nucleolar progressive degradation, which is temporally associated with cellular rRNA shutdown (Fig. 7f). Mimivirus infection also caused changes in nucleolus architecture, but such alterations were not comparable to those observed upon Tupanvirus infection (Fig. 7e, f). Although the formation of vesicles containing ribosomes and nuclear degradation can be related to cellular rRNA shutdown during Tupanvirus infection, the mechanisms involved in cellular rRNA degradation after the formation of large vesicles containing ribosomes remain to be investigated. One possibility that should be explored is that Tupanvirus might preferentially target some ribosomes to favor the translation of its own (as opposed to cellular) proteins, as previously described for poxviruses[27].

The toxicity effect and rRNA shutdown are independent of Tupanvirus replication; instead, they are caused by the viral particle (Fig. 8e–g). UV-light-inactivated particles continued to induce the depletion of *Acanthamoeba* rRNA (Fig. 8g). Although Tupanvirus is not able to replicate within *Tetrahymena* sp., it is phagocytosed in a voracious manner (as are other available macromolecular structures) and forms large intracellular vesicles, where the capsid and tail release their content inside the protist cytoplasm (Fig. 8i–k). The virus induces gradual vacuolization (Fig. 8i), loss of motility, a decrease in the phagocytosis rate

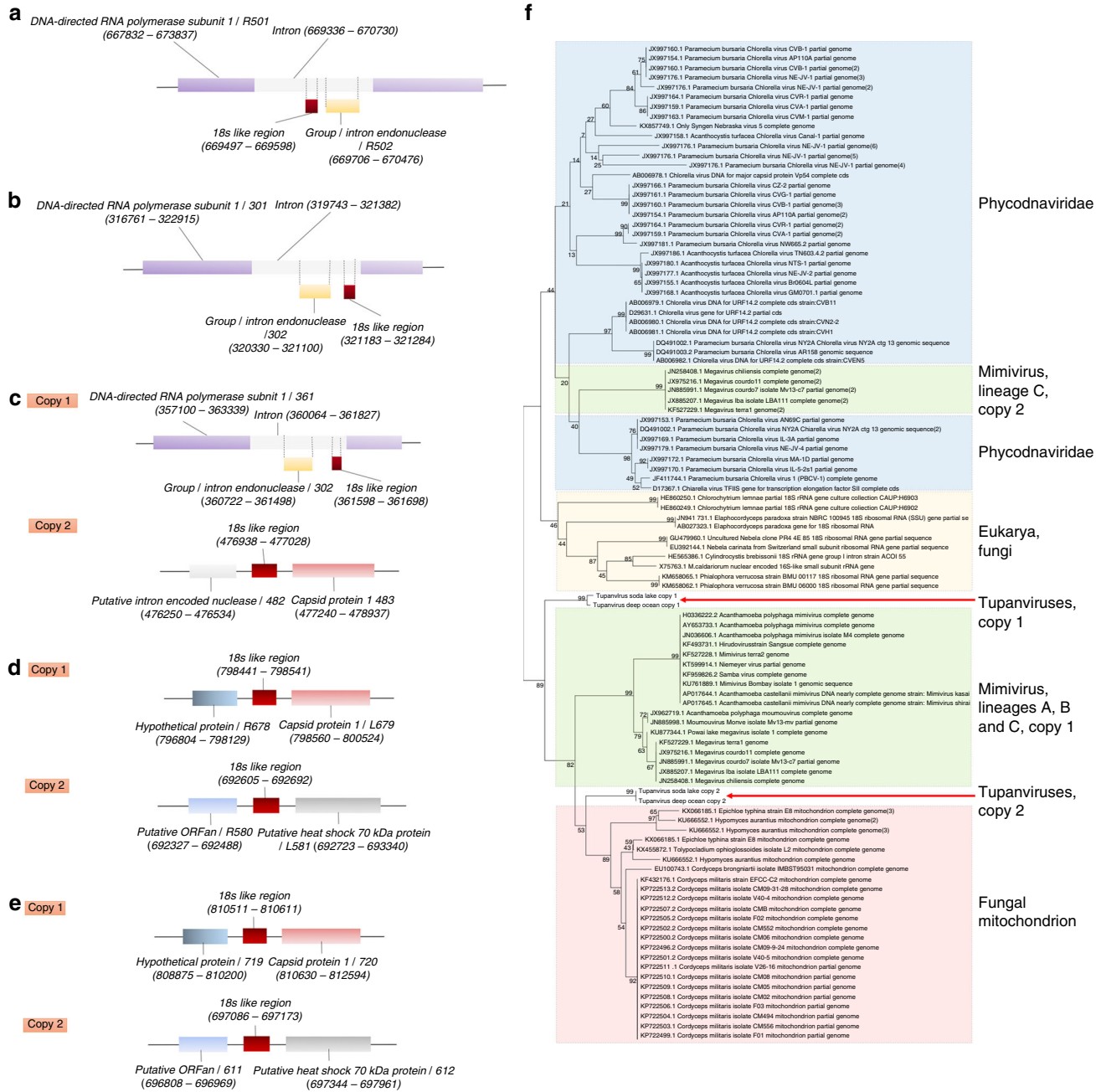

**Fig. 6** Adjacent regions of 18S rRNA intronic sequences in the genus *Mimivirus* and Tupanvirus and maximum likelihood phylogenetic tree of 18S rRNA intronic region. Core sequences are represented for lineages A (**a**), B (**b**), C (**c**), Tupanvirus soda lake (**d**) and Tupanvirus deep ocean (**e**). Phylogenetic tree of 18S rRNA intronic region present in mimivirus (**e**), *Phycodnaviridae*, eukaryotes and fungi mitochondrion

(Fig. 8m), a decrease in rRNA abundance (Fig. 8l) and triggers nuclear degradation (Fig. 7g), similar to the effects observed in *A. castellanii* cells with high multiplicities of infection (Fig. 8a–d). We performed in vitro simulations to determine whether this toxicity could affect the maintenance of Tupanvirus populations in a solution containing both *Tetrahymena* and *Acanthamoeba*. Our data suggest that at M.O.I. of 10 the reduction of the physiological activity of *Tetrahymena* sp., a (non-host) predator, decreases the ingestion of Tupanvirus particles (Fig. 8n), improving their chances to find its host, *Acanthamoeba*, and keeping viral titers constant in the system. In contrast, mimivirus particles, which do not present any toxicity to *Tetrahymena* sp., are quickly predated, and the virus is eliminated from the system after some days (Fig. 8n). Although we have no clues about the

hosts of tupanviruses at their original habitats nor if such high M. O.I. would be expected in nature, to our knowledge, a viral particle responsible for the modulation of host and non-host organisms independent of viral replication has not been previously described.

## Discussion

Considering that tupanviruses comprise a sister group to amoebal mimiviruses, we can hypothesize that the ancestors of these clades of *Mimiviridae* could had a more generalist lifestyle and were able to infect a wide variety of hosts. In this view, the ancestors of tupanviruses (and maybe of amoebal mimiviruses) might have already been giant viruses that underwent reductive evolution,

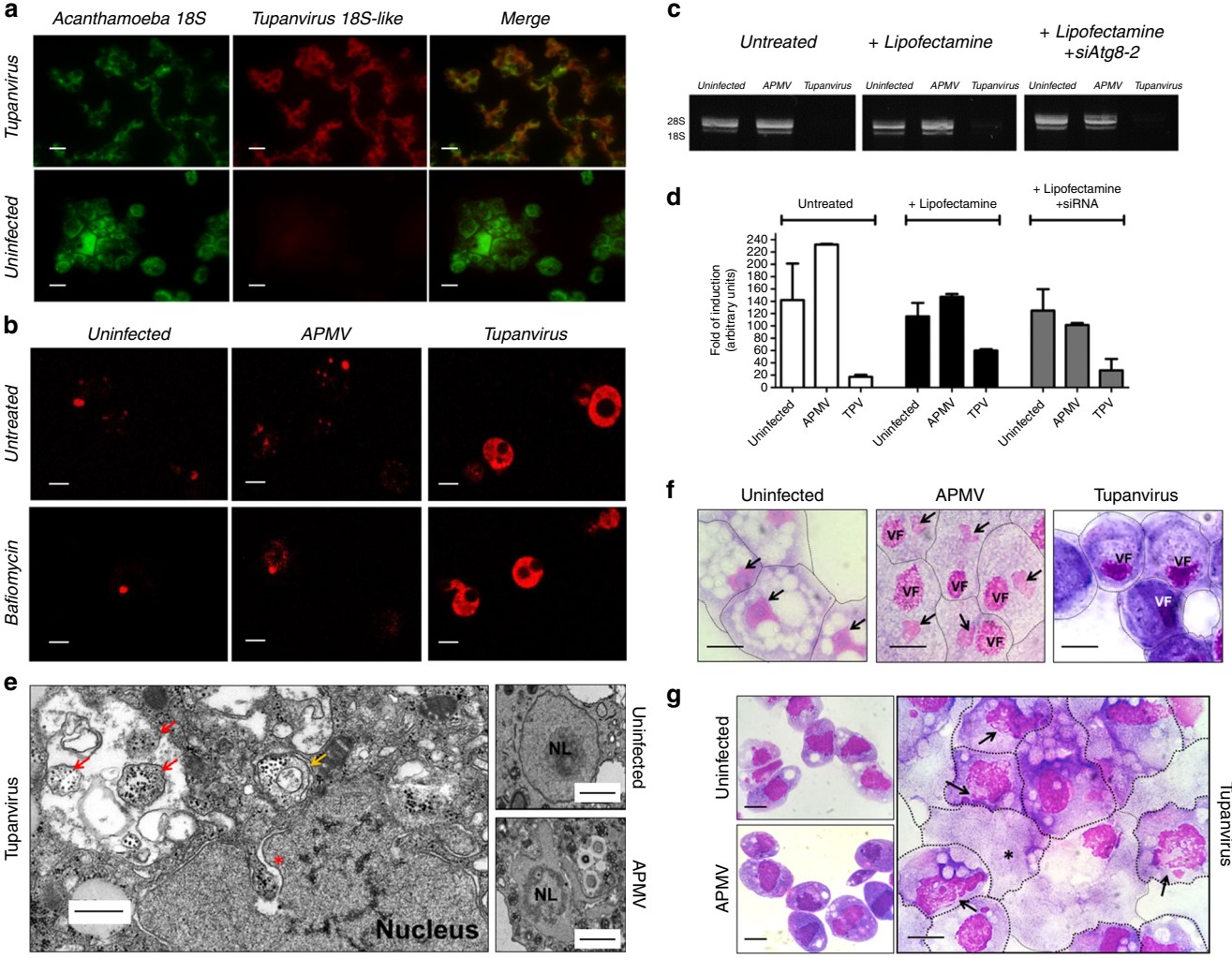

**Fig. 7** Tupanvirus soda lake biological features in a host (*A. castellanii*) and non-host (*Tetrahymena* sp.). **a** Expression of Tupanvirus 18S intronic sequence-copy 1 transcript, 12 h post-infection observed by fluorescence in situ hybridization (FISH) (red). Tupanvirus-induced shutdown of *A. castellanii* ribosomal RNA18S transcripts (green). Scale bars, 10 μm. **b** Tupanvirus, but not mimivirus, induces strong acidification of *A. castellanii* cytoplasm (9 h.p.i.), even in the presence of bafilomycin A1. Scale bars, 10 μm. **c**, **d** The silencing of the canonical autophagy protein Atg8-2 does not prevent rRNA shutdown induced by Tupanvirus infection. Error bars, standard deviation. **e** Electron microscopy of *A. castellanii* infected by Tupanvirus (2 h.p.i.), highlighting the degradation of the nucleolus, nuclear disorganization and the formation of ribosome-containing vesicles near nuclear membranes. Scale bar, 500 nm. Red arrow: single-membrane vesicles containing ribosomes; orange arrow: double-membraned vesicles containing ribosomes; asterisk: ribosomes wrapping by the external nuclear membrane. Right: electron microscopy of uninfected amoebae and APMV infected cell (8 h.p.i.) showing a mild nucleolar disorganization in the presence of the virus. Scale bars: 1 μm. **f** Haemacolour staining showing the nuclear degradation of *A. castellanii* induced by Tupanvirus infection (9 h.p.i.) compared with infection by mimivirus (APMV) and uninfected cells. Tupanvirus-infected cells are purplish because of cytoplasm acidification. Scale bars, 5 μm. Arrows: nucleus, when present; VF: viral factory. **g** Haemacolour staining showing the nuclear degradation in *Tetrahymena sp.* induced by Tupanvirus infection (4 days post-inoculation) compared with mimivirus (APMV)-inoculated cells and uninfected cells. Tupanvirus-infected cells present an atypical shape and intense vacuolization, and some cells lack a nucleus (asterisk). Arrows: nucleus under degradation. Scale bars, 5 μm. The experiments were performed 3 times independently, with two replicates each

although some genes could have been acquired over time, as previously hypothesized for other mimiviruses[5,23,28,29]. A reductive evolution pattern is typical among obligatory intracellular parasites[30–32]. In these cases, the organisms lose genes related to energy production, which is one of the main reasons for their obligatory parasitic lifestyle. In an alternative scenario, a simpler ancestor could had substantially acquired genes over time and became more resourceful, being able to infect a broader host range[5,33]. Nevertheless, tupanvirus presents the most complete translational apparatus among viruses, and its discovery takes us one step forward in understanding the evolutionary history of giant viruses.

## Methods

**Virus isolation and host-range determination**. In April 2014, a total of 12 sediment samples were collected from soda lakes in Southern Nhecolândia, Pantanal, Brazil. In 2012, 8 ocean sediments samples were collected from 3000 meters below water line surface at Bacia de Campos, in Campos dos Goytacazes, Rio de Janeiro, Brazil. The collection was performed by a submarine robot, during petroleum prospection studies performed by the Petrobras Company, and kindly provided to our group. The samples were stored at 4 °C until the inoculation process. The samples were transferred to 15 mL flasks, and 5 mL of Page's Amoebae Saline (PAS) was added. The solution was stored for 24 h to decant the sediment. The liquid was then subjected to a series of filtrations: first through paper filter and then through a 5 μm filter to remove large particles of sediment and to concentrate any giant viruses present. For co-culture, the cells used were *A. castellanii* (strain NEFF) and *V. vermiformis* (strain CDC 19), purchased from ATCC. These cell strains

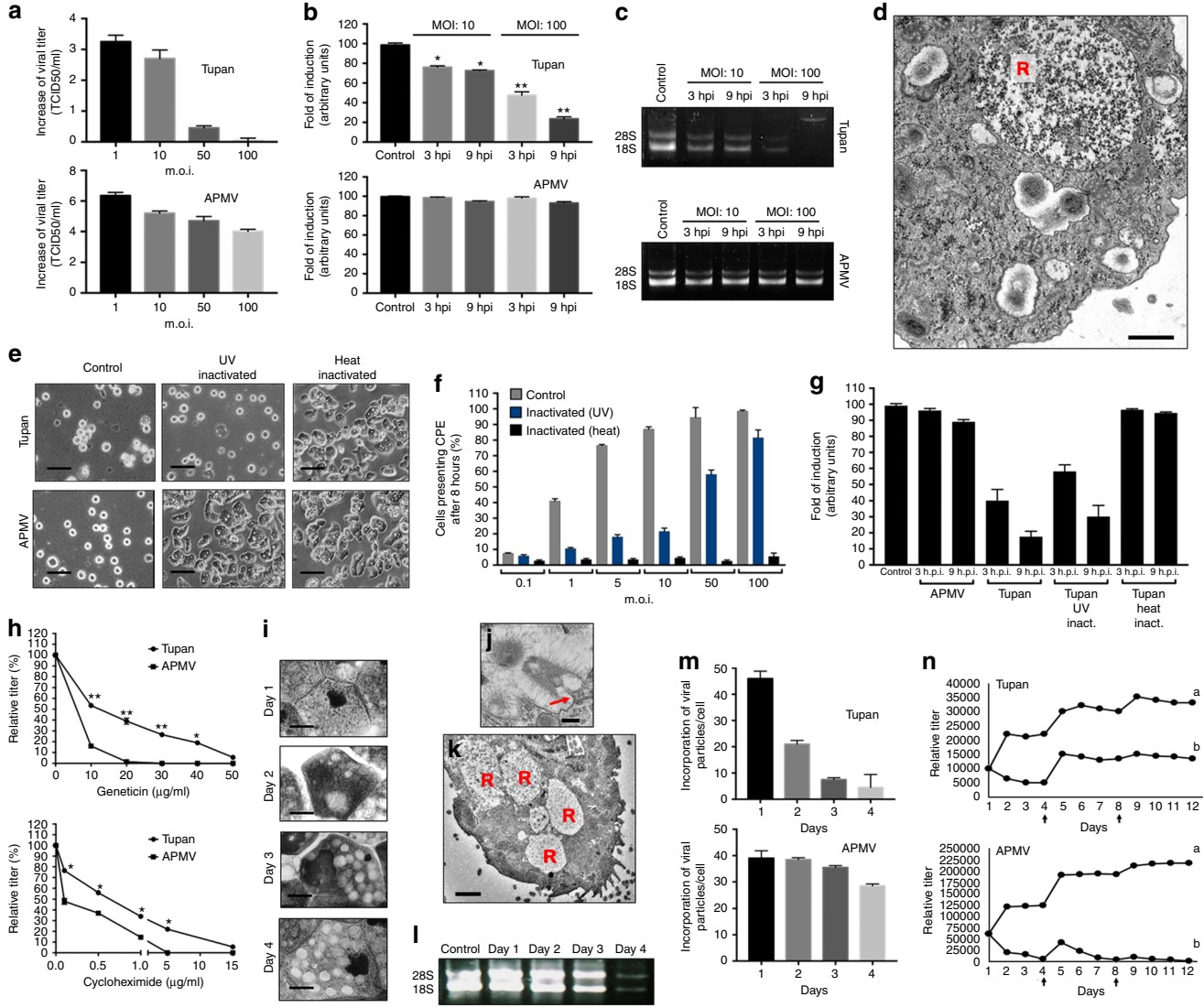

**Fig. 8** rRNA shutdown induced by Tupanvirus and toxicity assays. **a** Tupanvirus and mimivirus titers (log$_{10}$ values) 24 h.p.i. in *Acanthamoeba castellanii* at distinct MOIs. **b** Ribosomal 18S RNA relative measure by qPCR from *A. castellanii* infected by tupanvirus or mimivirus at an MOI of 10 or 100, 3 and 9 h post-infection. **c** Electrophoresis gel showing ribosomal 18S and 28S RNA from *A. castellanii* under the same conditions described in **b**. **d** Vesicle containing a large amount of *A. castellanii* ribosomes (R) 6 h after Tupanvirus infection. Scale bar, 1 μm. (**e**) Cytopathic effect of *A. castellanii* inoculated with Tupanvirus or mimivirus after UV or heat inactivation, MOI of 100, 8 h post-inoculation. Scale bar, 20 μm. **f** Counting of *A. castellanii* presenting cytopathic effect 8 h post-inoculation with tupanvirus inactivated by UV or heating under different MOIs. **g** Ribosomal 18S RNA relative measure by qPCR from *A. castellanii* infected by UV-inactivated or heat-inactivated Tupanvirus, or APMV, at an MOI of 100, 3 and 9 h post-infection. **h** Dose–response of Tupanvirus and APMV in *A. castellanii* pre-treated with distinct doses of geneticin or cycloheximide. (**i**) Progressive vacuolization of tetrahymena after infection with Tupanvirus (days 1–4). Scale bars, 10 μm. **j** Tupanvirus tail content releasing in tetrahymena 1 h post-inoculation (TEM). Scale bar, 200 nm. **k** Vesicles containing large amounts of *Tetrahymena* ribosomes (R) after Tupanvirus infection. Scale bar, 3.5 μm. **l** Electrophoresis gel showing rRNA shutdown in tetrahymena inoculated with Tupanvirus at an MOI of 10 (days 1–4). **m** Rate of particles incorporation per tetrahymena cell (days 1–4). **n** Simulations showing the decrease in APMV and maintenance of Tupanvirus populations over the analyzed days after infection of a mix of *Acanthamoeba castellanii* and tetrahymena at an MOI of 10 (lines indicated by 'b' in both graphs of **n**). At days 4 and 8, fresh PYG medium and 10$^5$ *A. castellanii* were added to the systems (arrows). For the negative control of this experiment we pre-treated tetrahymena with 20 μg/ml of geneticin (lines indicated by 'a' in both graphs of **n**). In this case, both APMV and Tupanvirus were able to grow in the system. Statistical analyses in **b** and **h**: t-test based on control groups (**b**) or corresponding virus/drug concentrations (**h**). *:$p < 0.05$; **:$p < 0.01$. The experiments were performed 3 times independently, with two replicates each. Error bars (**a**, **b**, **f**, **g**, **h**, **m** and **n**), standard deviation

were stored in 75 cm$^2$ cell culture flasks containing 30 mL of peptone yeast extract glucose medium (PYG) at 28 °C. After 24 h of growth, cells were harvested and pelleted by centrifugation. The supernatant was removed, and the amoebae were resuspended three times in sterile PAS. After the third washing, 500,000 *A. castellanii* or *V. vermiformis* were resuspended in PAS or TS solutions and seeded in 24-well plates. The amoebae suspensions were added to an antibiotic mix containing ciprofloxacin (20 μg/mL; Panpharma, Z.I., Clairay, France), vancomycin (10 μg/mL; Mylan, Saint-Priest, France), imipenem (10 μg/mL; Mylan, Saint-Priest, France), doxycycline (20 μg/mL; Mylan, Saint-Priest, France), and voriconazole (20

μg/mL; Mylan, Saint-Priest, France). Each 100 μL of sample was mixed and inoculated in the numbered (1–12) wells and incubated at 30 °C in a humid chamber. A negative control was used in each plate. The wells were observed daily under an optical microscope. After 3 days, new passages of the inoculated wells were performed in the same manner until the third passage. In this passage, the content of the wells presenting lysis and cytopathic effects were collected and stored for production and analysis of the possible isolates by haemacolour staining and electron microscopy using the negative stain technique. Of the twelve tested samples from Pantanal, we found Tupanvirus (soda lake) in three. In only one

ocean sample we did isolate Tupanvirus (deep ocean). To evaluate the Tupanvirus soda lake host range, a panel of cell lines was subjected to virus infection at an multiplicity of infection (MOI) of 5: *Acanthamoeba castellanii* (ATCC30010), *Acanthamoeba royreba* (ATCC30884), *Acanthamoeba griffin* (ATCC50702), *Acanthamoeba* sp. E4 (IHU isolate), *Acanthamoeba* sp. Micheline (IHU isolate), *Vermamoeba vermiformis* (ATCC50237), *Dictyostelium discoideum* (ATCC44841), *Willartia magna* (ATCC50035), *Tetrahymena hyperangularis* (ATCC 50254), *Trichomonas tenax* (ATCC 30207), RAW264.7 (Mouse leukemic monocyte-macrophage) (ATCCTIB71) and THP-1 (human monocytic cell line) (ATCCTIB202). Cell lines were tested for mycoplasma. The assays were carried out in 24-well plates, and cells were incubated for 24 or 48 h. The tupanvirus titer was measured in *A. castellanii* by end-point and calculated by the Reed-Muench method[34]. In parallel, the samples were subjected to qPCR, targeting the tupanvirus tyrosyl RNA synthetase (5′-CGCAATGTGTGGAGCCTTTC-3′ and 5′-CCAAGA-GATCCGGCGTAGTC-3′) and aiming to verify viral genome replication (Biorad, CA, USA). Tupanvirus was propagated in 20 *A. castellanii* 175 cm$^2$ cell culture flasks in 50 mL PYG medium. The particles were purified by centrifugation through a sucrose cushion (50%), suspended in PAS and stored at −80 °C. Purified particles were used for genome sequencing, proteomic analysis[10], and microscopic and biological assays.

**Cycle and virion characterization.** All biological tests were performed with Tupanvirus soda lake only. To investigate the viral replication cycle by TEM, 25 cm$^2$ cell culture flasks were filled with $10 \times 10^6$ *A. castellanii* per flask, infected by Tupanvirus at a multiplicity of infection of 10 and incubated at 30 °C for 0, 2, 4, 6, 8, 12, 15, 18, and 24 h. One hour after virus-cell incubation, the amoeba monolayer was washed three times with PAS buffer to eliminate non-internalized viruses. A total of 10 mL of the infected cultures was distributed into new culture flasks. A culture flask containing only amoeba was used as the negative control. The infected cells and control sample were fixed and prepared for electron microscopy[14]. For immunofluorescence, *A. castellanii* cells were grown, infected by Tupanvirus at a multiplicity of infection of 1 as described and added to coverslips for 0, 2, 4, 6, 8, 12, 15, 18, and 24 h. After infection, the cells were rinsed in cold phosphate-buffered saline (PBS) and fixed with 4% paraformaldehyde (PFA) in PAS for 10 mins. After fixation, cells were permeabilized with 0.2% Triton X-100 in 3% bovine serum albumin (BSA)–PAS for 5 min, followed by rinsing with 3% BSA–PBS three times. Cells were then stained for 1 h at room temperature with an anti-tupanvirus antibody produced in mice (According Aix Marseille University ethics committee rules). After incubation with an anti-mouse secondary antibody, fluorescently labeled cells were viewed using a Leica DMI600b microscope. For Tupanvirus virion characterization, we also used scanning electron microscopy (SEM)[35]. Chemical treatment with proteases and sonication was performed as described elsewhere[35] to investigate fiber composition and the attachment between capsid and tail. For tomography videos, tilt series were acquired on a Tecnai G2 transmission electron microscope (FEI) operated at 200 keV and equipped with a 4096 × 4096 pixel resolution Eagle camera (FEI) and Explore 3D (FEI) software. The tilt range was 100°, scanned in 1° increments. The magnification ranged between 6,500 and 25,000, corresponding to pixel sizes between 1.64 and 0.45 nm, respectively. The image size was 4,096 × 4,096 pixels. The average thickness of the obtained tomograms was 298 ± 131 nm (*n* = 11). The tilt series were aligned using ETomo from the IMOD software package (University of Colorado, USA) by cross-correlation (http://bio3d.colorado.edu/imod/). The tomograms were reconstructed using the weighted-back projection algorithm in ETomo from IMOD. ImageJ software was used for image processing.

**Genomes sequencing and analyses.** The tupanviruses genomes were sequenced using an Illumina MiSeq instrument (Illumina Inc., San Diego, CA, USA) with the paired end application. The sequence reads were assembled de novo using ABYSS software and SPADES, and the resulting contigs were ordered by the Python-based CONTIGuator.py software. The obtained draft genomes were mapped back to verify the read assembly and close gaps. The best assembled genome was retained, and the few remaining gaps (three) were closed by Sanger sequencing. The gene predictions were performed using the RAST (Rapid Annotation using Subsystem Technology) and GeneMarkS tools. Transfer RNA (tRNA) sequences were identified using the ARAGORN tool. The functional annotations were inferred by BLAST searches against the GenBank NCBI non-redundant protein sequence database (nr) (*e*-value $< 1 \times 10^{-3}$) and by searching specialized databases through the Blast2GO platform. Finally, the genome annotation was manually revised and curated. The predicted ORFs that were smaller than 50 amino acids and had no hits in any database were ruled out. Tupanvirus codon and aa usages were compared with those of *A. castellanii* and other lineages of mimiviruses. Sequences were obtained from NCBI GenBank and subjected to CGUA (General Codon Usage Analysis). The global distribution of Tupanvirus tRNAs was analyzed and compared manually with viral aa usage considering the corresponding canonical codons related to each aa. Phylogenetic analyses were carried out based on the separate alignments of several genes, including family B DNA polymerase, 18 S rRNA intronic regions (copies 1/2) and 20 aminoacyl-tRNA synthetases (aaRS). The predicted aa sequences were obtained from NCBI GenBank and aligned using Clustal W in the Mega 7.0 software program. Trees were constructed using the maximum likelihood evolution method and 1000 replicates. The analysis of aaRS

domains was carried out using NCBI Conserved Domain Search (https://www.ncbi.nlm.nih.gov/Structure/cdd/wrpsb.cgi). A search for promoter sequences was performed in intergenic regions based on a search for the mimivirus canonical AAAATTGA promoter sequence, as previously described[20]. Single-nucleotide polymorphisms (SNP) in the AAAATTGA promoter sequence were also considered for each base, considering all possibilities. Gene sets available for members of the family *Mimiviridae* and those of Tupanvirus soda lake and Tupanvirus deep ocean were used for analyses of the mimivirus pangenome. Groups of orthologues were determined using the Proteinortho tool V51 with $1e-3$ and 50% as the *e*-value and coverage thresholds, respectively. Concurrently, BLAST searches were performed using ORFs of all mimivirus genomes available in the NCBI GenBank nucleotide sequence database against the set of clusters of orthologous groups previously delineated for mimiviruses (mimiCOGs) (*n* = 898), with $1e-3$ and 50% as the *e*-value and coverage thresholds, respectively. For rhizome preparations, all coding sequences were blasted against the NR database, and results were filtered to retain the best hits. Taxonomic affiliation was retrieved from NCBI. For the construction of a translation-associated elements network, the different classes of translation elements of each organism included in the analysis were obtained by searching for each component within their genome, according to protein function annotation using Blastp searches against the GenBank NCBI non-redundant protein sequence database. The tRNA components were obtained using the ARAGORN tool. Different tRNA molecules were included in the analysis, considering the anti-codon sequence. Repeated elements were eliminated to avoid analysis of duplicate events. The layout of the network was generated by a force-directed algorithm—followed by local rearrangement for visual clarity, leaving the network's overall layout unperturbed—using the program Gephi (https://gephi.org).

**Ribosomal RNA shutdown and toxicity assays.** To investigate the toxicity of Tupanvirus particles, 1 million *A. castellanii* cells were infected with Tupanvirus or mimivirus at a multiplicity of infection of 1, 10, 50, or 100 and incubated at 32 °C. At 0 and 24 h post-infection, the cell suspensions were collected and titered as previously described. A fraction of this suspension (200 μL) was subjected to RNA extraction (Qiagen RNA extraction Kit, Hilden, Germany). The RNA was subjected to reverse transcription by using Vilo enzymes (Invitrogen, CA, USA) and then used as a template in qPCR targeting *A. castellanii* 18 S rRNA (5′-TCCAATTTTCTGCCACCGAA-3′ and 5′-ATCATTACCCTAGTCCTCGCG C-3′). The values were expressed as arbitrary units (delta-Ct). Normalized amounts of the original RNA extracted from each sample were electrophoresed in 1% agarose gel with TBE buffer and run at 150 V. TEM over the entire testing period was performed to evaluate the presence of ribosome-containing vesicles and other cytological alterations. To investigate the nature of virion toxicity, purified Tupanvirus was inactivated by UV light (1 h of exposure, 60 W/m$^2$) or heating (80 °C, 1 h)—inactivation was confirmed by inoculation on *Acanthamoeba castellanii*, CPE was observed for 5 days and lack of replication was confirmed by qPCR—and inoculated onto *A. castellanii* containing 500,000 cells at multiplicities of infection of 0.1, 1, 5, 10, 50, and 100. The assays were performed in PAS solution. The cytopathic effect was documented and quantified in a counting-cells chamber. Inactivated mimivirus was used for comparison. To determine whether Tupanvirus-induced shutdown of amoebal 18 S rRNA even after inactivation, 500,000 cells were infected (at a multiplicity of infection of 100) and collected at 3 and 9 h post-infection, and amoebal 18 S rRNA levels were measured by qPCR. APMV was used as the control. The sensitivity of Tupanvirus and mimivirus to the translation-inhibiting drugs geneticin and cycloheximide was tested. A total of 500,000 *A. castellanii* cells were pre-treated with different concentrations of the drugs (0–50 and 0–15 μg/ml, respectively) for 8 h and then infected at a multiplicity of infection of 10. Twenty-four hour post-infection, cells were collected, and the viral titers were measured. To investigate the toxicity effect of Tupanvirus particles in the non-host *Tetrahymena* sp., 1 million fresh cells were infected at a multiplicity of infection of 10 in a medium composed of 50% PYG and 50% PAS. The cytopathic effect was monitored for 4 days post-infection, given the reduction of cell movement and vacuolization (lysis or viral replication was not observed). Each day post-infection, 100 μL of infected cell suspension was collected and subjected to cytospin and haemacolour staining to observe vacuolization and other cytological alterations induced by the virus. Other 100 μL aliquots were used to investigate the occurrence of rRNA shutdown induced by Tupanvirus. To this end, the samples were subjected to RNA extraction and electrophoresis. Viral infection in *Tetrahymena* sp. was also observed by TEM at a multiplicity of infection of 10. To determine whether Tupanvirus particles affect the rate of *Tetrahymena* sp. phagocytosis because of toxicity, the rate of viral particle incorporation per cell was calculated during the period of infection. The ratio of TCID50 (infectious entities) and total particles was first calculated by counting the number of viral particles in a counting chamber (approximately 1 TCID$_{50}$ to 63 total particles). One million *Tetrahymena* cells were infected by Tupanvirus or mimivirus at an MOI of 10 TCID$_{50}$. Twelve hour post-infection, the number of viral particles in the medium was estimated by counting the remaining (non-phagocytized) particles. An input of 10 TCID$_{50}$ per cell was added each day post-infection (in separate flasks, one for each day), and the rate of particles phagocytosis was calculated 12 h post-input. For the calculation, the remaining particles from the day before were considered (counted immediately before the input). Considering the toxicity caused by Tupanvirus, but not APMV, in tetrahymena, we conducted an in vitro experiment

aiming to investigate the ability to maintain Tupanvirus or APMV in a system containing both *Acanthamoeba* (host) and *Tetrahymena* (non-host, predator of particles). Thus, *A. castellanii* (900,000 cells) and *Tetrahymena* (100,000 cells) were added simultaneously to the same flask, then infected by Tupanvirus or mimivirus at an MOI of 10 and observed for 12 days. One flask per observation day was prepared. At days 4 and 8, we added 500 μL of fresh medium (50% PYG and 50% PAS) and 100,000 *A. castellanii*, the permissive host. Each day post-infection, the corresponding flask was collected and subjected to titration as previously described. The same experiment was carried out by pre-treating *Tetrahymena* (8 h before infection) with 20 μg/ml of geneticin as negative controls.

**Analysis involving Tupanvirus 18 S rRNA intronic region.** All analyses involving the genomic environment of copies 1 and 2 were conducted based on the annotation of Tupanvirus. In the best-hits evaluation, the core sequences of copies 1 and 2 were used for nucleotide BLAST analysis using blastn. The resulting 100 best hits were tabulated, and the information was used to construct diagrams. For the phylogenetic analysis, the sequences of these best hits were also aligned, using Clustal W in the Mega 7.0 software, and constructed using the maximum likelihood evolution method of 1,000 replicates. To analyze subjacent regions of the core sequence of 18 S rRNA intronic regions in the *Mimiviridae* family, one member of the lineages A (HQ336222.2), B (JX962719.1) and C (JX885207.1) was chosen and analyzed. The expression of both copies was checked using fluorescence in situ hybridization (FISH) and qPCR. For this, *A. castellanii* cells were infected with Tupanvirus with a multiplicity of infection of 5 and collected at 30 min and at 6 and 12 h post-infection. As a control, *A. castellanii* cells were also incubated with PAS alone and collected. At the indicated times, cells and the supernatant were collected and centrifuged at $800 \times g$ for 10 min. For FISH, the pellet was resuspended in 200 μL of PAS, submitted to cytospin and the cells were fixed in cold methanol for 5 min. Specific probes targeting the 18 S rRNA of *A. castellanii* (5′-TTCACGGTAAACGATCTGGGCC-3′-fluorophore Alexa 488), copy 1 RNA (5′-AGTGGAACTCGGGTATGGTAAAA-3′-fluorophore Alexa 555) and copy 2 RNA (5′-GGCCAAGCTAATCACTTGGG-3′-fluorophore Alexa 555) were diluted and applied at 2 μM in hybridization buffer (900 mM NaCl, 20 mM Tris/HCL, 5 mM EDTA, 0.01% SDS, 10–25% deionized-formamide in distilled-H$_2$O). The hybridization buffer containing the probes was added to the slides and the hybridization was carried out at 46 °C overnight in a programmable temperature-controlled slide-processing system (ThermoBrite StatSpin, IL, USA). Post-hybridization washes consisted of 0.45–0.15 M NaCl, 20 mM Tris/HCL, 5 mM EDTA, and 0.01% SDS at 48 °C for 10 min. Slides were analyzed using a DMI6000B inverted research microscope (Leica, Wetzlar, Germany). To qPCR the pellet of infected cells was also washed with PAS and then used for total RNA extraction using the RNeasy mini kit (Qiagen, Venlo, Netherlands). The extracted RNA was treated with the Turbo DNA-free kit (Invitrogen, CA, USA) and then used as a template in reverse transcription (RT) reactions carried out using SuperScript Vilo (Invitrogen, CA, USA). The resultant cDNA was used as a template for quantitative real-time PCR assays using the QuantiTect SYBr Green PCR Kit (Qiagen RNA extraction Kit, Hilden, Germany) and targeting copies 1 (primers 5′-GCATCAAGTGCCAACCCATC-3′ and 5′-CTGAAATGGGCAATCCGCAG-3′) and 2 (primers 5′-CCAAGTGATTAGCTTGGCCATAA-3′ and 5′-CGGGAAGTCCCTAAAGCTCC-3′) of the intergenic18S rRNA region in TPV. To normalize the results, primers targeting the GAPDH housekeeping gene of *Acanthamoeba* (primers 5′-GTCTCCGTCGTCGATCTCAC-3′ and 5′-GCGGCCTTAATCTCGTCGTA-3′) were also used. qPCR assays were performed in a BioRad Real-Time PCR Detection System (BioRad) using the following thermal conditions for all genes: 15 min of pre-incubation at 95 °C followed by 40 amplification cycles of 30 s at 95 °C, 30 s at 60 °C and 30 s at 72 °C. The results were analyzed using the relative quantification methodology of $2^{(-\Delta\Delta ct)}$.

**Investigation of the nature of ribosomal RNA shutdown.** To investigate the shutting down of the host rRNA and verify whether this phenomenon was related to the canonical ribophagy/autophagy process, tests using two acidification and lysosome-vesicle fusion inhibitors (chloroquine and bafilomycin A) were performed. The pH of infected cells and the effect of Atg8-2 silencing on shutdown were also tested. For the inhibitor assays, $5 \times 10^5$ *A. castellanii* cells cultured in PYG medium were infected with Tupanvirus or mimivirus at a multiplicity of infection of 100 and incubated at 32 °C. At 1 h post-infection, chloroquine (Sigma-Aldrich, MO, USA) at a final concentration of 100 μM or bafilomycin A (Sigma-Aldrich, MO, USA) at a final concentration of 10 nM was added to the infected cell suspensions. As a control, *A. castellanii* cells not infected were also treated with these inhibitors under the same conditions. After 3 and 9 h post-infection, cells and the supernatant were collected and centrifuged at $800 \times g$ for 10 min. The supernatant was discarded, and the pellet was submitted to RNA extraction (Qiagen RNA extraction Kit, Hilden, Germany). From the extracted RNA, 10 μL of each sample was electrophoresed in 1.5% agarose gel with TBE buffer and run at 135 V, and 14 μL was submitted to reverse transcription to measure the amoebal 18 S rRNA levels by qPCR as previously described. To investigate the acidification caused by Tupanvirus or mimivirus infection, *A. castellanii* cells were also submitted to the same pattern of infection and treatment with bafilomycin A, as previously described. In addition, 1 h before the collection time, the cells were incubated with LysoTracker Red DND-99 (Thermo Fisher Scientific, Massachusetts, United States)

at a final concentration of 75 nM. After 9 h post-infection, cells and the supernatant were collected and centrifuged at $800 \times g$ for 10 min. The supernatant was discarded, and the pellet was resuspended in 1 mL of PAS medium containing only bafilomycin A (10 nM). A total of 20 μL of this suspension was added to glass slides and cover slipped. Analyses were performed using a confocal microscope (Zeiss, Jena, Germany). A small interfering RNA (siRNA) targeting the Atg8-2 gene of *A. castellanii* was synthesized by Eurogentec (Liège, Belgium) based on the cDNA sequence of the gene. The siRNA duplex with sense (5′-GAACUCAUGUCGCACAUCUU-3′) and anti-sense (5′-AGAUGUGCGACAUGAGUUCUU-3′) sequences was used. The siRNA tagged with a fluorescence dye was transfected onto *A. castellanii* trophozoites at a density of $1 \times 10^6$ cells. The control of transfection was performed using fluorescence microscopy. The biological effect of siRNA was check by qPCR and by the observation of the inhibition of acanthamoebal encystment, which is dependent on Atg8-2. Finally, modifications of *A. castellanii* nucleus/nucleolus structure after infection with Tupanvirus and mimivirus were investigated. A total of $10^6$ cells were infected with Tupanvirus or mimivirus at an MOI of 10, stained by haemacolour and treated with SYTO RNASelect Green Fluorescent cell stain (Invitrogen, USA) following the manufacturer's instructions. After 9 h.p.i., cells were observed under an immunofluorescence microscope to observe modifications to the nucleus/nucleolus of infected and control cells. In parallel, this preparation was submitted to electron microscopy.

**Data availability.** The Tupanvirus genome sequences have been deposited in GenBank under accession codes KY523104 (soda lake) and MF405918 (deep ocean). Proteomic data have been deposited in PRIDE archive under accession code PXD007583. All other data supporting the findings of this study are available within the article and its Supplementary Information, or from the corresponding author upon reasonable request.

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

## Acknowledgements

We thank our colleagues from URMITE and Laboratório de Vírus of Universidade Federal de Minas Gerais for their assistance, particularly Julien Andreani, Jean-Pierre Baudoin, Gilles Audoly, Amina Cherif Louazani, Lina Barrassi, Priscilla Jardot, Eric Chabrières, Philippe Decloquement, Nicholas Armstrong, Said Azza, Emeline Baptiste, Claudio Bonjardim, Paulo Ferreira, Giliane Trindade and Betania Drumond. In addition, we thank the Méditerranée Infection Foundation, Centro de Microscopia da UFMG, CNPq (Conselho Nacional de Desenvolvimento Científico e Tecnológico), CAPES (Coordenação de Aperfeiçoamento de Pessoal de Nível Superior) and FAPEMIG (Fundação de Amparo à Pesquisa do estado de Minas Gerais) for their financial support. We thank Petrobras for the collection of sediments from ocean. This work was also supported by the French Government under the « Investissements d'avenir » (Investments for the Future) program managed by the Agence Nationale de la Recherche (ANR, fr: National Agency for Research), (reference: Méditerranée Infection 10-IAHU-03). J.A., B.R. and E.K. are CNPq researchers. B.L.S., J.A., L.S., P.C., and E.G.K. are members of a CAPES-COFECUB project.

## Author contributions

D.R., B.L.S., J.S.A., A.L., P.C., E.G.K, and E.G. designed the study and experiments. L.S., J.S.A, J.B.K., R.R., L.S., L.S.S., T.A., P.C., F.A. P.B., M.A., I.B., B.R., A.L., and H.S. performed sample collection, virus isolation, experiments and/or analyses. D.R., B.L.S., A.L., J.S.A., P.C., G.K., R.R., L.S., and L.S.S. wrote the manuscript. All authors approved the final manuscript.

## Additional information

**Competing interests:** The authors declare no competing financial interests.

