## [Peer Review File · Nature Communications]

Reviewers' comments:

Reviewer #4 (Remarks to the Author):

Reviewer: Hiroyuki Ogata (Professor, Kyoto University)

The authors describe a striking discovery of a new group of mimivirus relatives, which show several unique features distinct from previously isolated mimiviruses. The new viruses called Tupanviruses exhibit thick/long tails unlike other mimiviruses, as well as genomes more enriched in translation related genes than other relatives. Furthermore, Tupanviruses show a toxic effect in host as well as non-host predator organisms. The toxicity on predators of viruses raises a possibility, as the authors mention, that these viruses can modulate non-host organisms to increase viral survival chances in nature. Although, in this manuscript, this ecological scenario is not supported by evidences from natural environments, this recalls me the notion of Extended Phenotype coined by Richard Dawkins and is intriguing. Along with the cytotoxicity, the authors observed ribosomal shutdown apparently caused by the virion of Tupanviruses. Overall, this work is adequately comprehensive as an initial characterization of such viruses, and will open new windows for bioinformaticians and experimental virologists to investigate the evolutionary pathway to giant viruses as well as their ecological strategies for survival. As a genomics/bioinformatics researcher, I would like to give a few comments on the current version of the manuscript, while I would like to leave the evaluation of ribosomal shutdown aspect for other specialists.

The statement about translation related genes in the abstract that "even larger than some prokaryotes and eukaryotes" is misleading and a sort of overstatement, as authors restricted their comparison between the viruses and cellular organisms only on shared translation categories, from which ribosomal genes were excluded.

The authors do not provide detailed phylogenetic analysis of aaRSs, although the authors message (i.e., "it is not possible to state that the origin of most of these tupanvirus aaRS genes is cellular.") contradicts with the recent report on Klosneuviruses (Schulz et al., Science, 2017). I agree that such analysis requires time and could be left as a future study, but at least such a contradiction with previous related report should be stated.

The authors hypothesized that the group of Mimiviridae experienced a reductive evolution from their last common ancestor. Reductive evolution for Mimiviridae has been previously proposed along with the genome sequence analysis of related Megavirus (Arslan et al., PNAS, 2011) with lesser amount but based on, fairly, a similar type of data as in this study. I strongly propose that the authors cite Arslan et al or state in the manuscript differences in their and Arslans et al proposals.

Line 70: Fig. 1A,F,I: I did not find Fig. 1I.

Line 122: 25.1% is not "nearly half", and 11/87 does not correspond to the number of ORFans, which is stated to be "eight". I am confused.

Line 123 SI 3: This Table SI 3 is better if it describes gene function annotations. In addition, I do not understand the meaning of blank separators in the table.

Line 123: The sentence "Although 62 Tupanvirus virion proteins were not found by proteomics, either in Mimivirus or in Cafeteria roenbergensis virus particles, there are no distinct clues about which protein(s) could be associated with the tupanvirus fibrils and tail structure." needs rewording because it is impossible that Tupanvirus virion proteins are found in Mimivirus or in CroV particles. A word such as "homologs" may be missing here.

Reviewer #5 (Remarks to the Author):

This manuscript reports the discovery and characterization of two strains of tupanvirus, a giant dsDNA virus that is related to mimiviruses but presents several previously unseen features. The most striking difference to mimiviruses is the presence of a long, fiber-coated tail that appears to be hollow. Tupanvirus is also the largest mimivirus discovered so far, with a ~1.5 Mb dsDNA genome and 775 genes unique not found in other mimiviruses. The translation gene set of these viruses is impressive and suggests that tupanviruses are highly efficient in taking over host cell protein biosynthesis. The authors also report a cytotoxicity of the viral particles, even in non-permissive host cells such as Tetrahymena. This study is highly fascinating, reports novel findings, and is written in a comprehensible manner, although language editing is recommended. As described below, the discussion could be modified to include alternative evolutionary scenarios.

- Abstract, page 2. "Translation is the canonical frontier between the cell world and the virosphere" – Would the authors not consider energy production an equally important frontier?
- Page 4. The authors compare the viral factory to a nucleus. Can this statement be specified in more detail, i.e. what makes the viral factory nucleus-like?
- Page 5. This reviewer doubts that readers are familiar with the term "rhizome", and even though a reference is given, the term should be explained.
- Conclusions, page 11. Regarding the "more generalist lifestyle" and the "reductive evolution" of the mimivirus ancestor, why do the authors insist on such a lopsided interpretation? Is it not equally likely that the mimivirus ancestor was in the process of acquiring genes (including translation genes), which proved advantageous to its propagation, and that the tupanvirus lineage further extended its host range and continued to incorporate additional genes, whereas the mimiviruses described so far had a narrower host range, for which a not-so-extensive translational gene complement was sufficient? Also, would tupanvirus no longer be placed in the virosphere if it had ribosomes?

Reviewers' comments:

Reviewer #4 (Remarks to the Author):

Reviewer: Hiroyuki Ogata (Professor, Kyoto University)

The authors describe a striking discovery of a new group of mimivirus relatives, which show several unique features distinct from previously isolated mimiviruses. The new viruses called Tupanviruses exhibit thick/long tails unlike other mimiviruses, as well as genomes more enriched in translation related genes than other relatives. Furthermore, Tupanviruses show a toxic effect in host as well as non-host predator organisms. The toxicity on predators of viruses raises a possibility, as the authors mention, that these viruses can modulate non-host organisms to increase viral survival chances in nature. Although, in this manuscript, this ecological scenario is not supported by evidences from natural environments, this recalls me the notion of Extended Phenotype coined by Richard Dawkins and is intriguing. Along with the cytotoxicity, the authors observed ribosomal shutdown apparently caused by the virion of Tupanviruses. Overall, this work is adequately comprehensive as an initial characterization of such viruses, and will open new windows for bioinformaticians and experimental virologists to investigate the evolutionary pathway to giant viruses as well as their ecological strategies for survival. As a genomics/bioinformatics researcher, I would like to give a few comments on the current version of the manuscript, while I would like to leave the evaluation of ribosomal shutdown aspect for other specialists.

Answer: Dear Prof. Ogata, thank you very much for your comments and suggestions.

The statement about translation related genes in the abstract that “even larger than some prokaryotes and eukaryotes” is misleading and a sort of overstatement, as authors restricted their comparison between the viruses and cellular organisms only on shared translation categories, from which ribosomal genes were excluded.

Answer: As suggested, we modified this statement to make clear that ribosomal proteins were not considered in this analysis.

Abstract: *“Remarkably, these giant viruses present the largest translational apparatus within the virosphere, even larger than some prokaryotes and eukaryotes (not considering ribosomal proteins)1-3, with up to 70 tRNA and a gene set comprising 20 aaRS, 11 factors for all translation steps, and factors related to tRNA/mRNA maturation and ribosome protein modification.”*

Main text: *The comparison between contents in translation-related categories of genes present in tupanviruses and cellular organisms reveals that tupanviruses present a richer gene set than Candidatus Carsonella ruddii (Bacteria) and Nanoarchaeum equitans (Archaea) (not considering ribosomal proteins).*

The authors do not provide detailed phylogenetic analysis of aaRSs, although the authors message (i.e., “it is not possible to state that the origin of most of these tupanvirus aaRS genes is cellular.”) contradicts with the recent report on Klosneuviruses (Schulz et al., Science, 2017). I agree that such analysis requires time and could be left as a future study, but at least such a contradiction with previous related report should be stated.

Answer: we agree that point must be addressed. The following sentence (yellow) was added in the manuscript.

“Based on the 20 aaRS trees, it is not possible to state that the origin of most of these tupanvirus aaRS genes is cellular (SI 8). This contrasting result to that reported by Schultz et al, 2017 for klosneuviruses may be explained by the different sampling used for alignments and trees construction²⁴”

In order to highlight the difficulty to determine the origin of most of these tupanvirus aaRS genes, we reconstructed the rhizome at the sequence level of one aaRS gene (glutaminyl-tRNA synthetase as an example). These analyses revealed the mosaic structure of such a gene (SI 10).

The authors hypothesized that the group of Mimiviridae experienced a reductive evolution from their last common ancestor. Reductive evolution for Mimiviridae has been previously proposed along with the genome sequence analysis of related Megavirus (Arslan et al., PNAS, 2011) with lesser amount but based on, fairly, a similar type of data as in this study. I strongly propose that the authors cite Arslan et al or state in the manuscript differences in their and Arslans et al proposals.

Answer: as suggested, we have inserted this reference (number 29).

Line 70: Fig. 1A,F,I: I did not find Fig. 1I.

Answer: sorry for the gap. It was corrected:

“Microscopic analysis suggests that the capsid and tail are not tightly attached (Fig. 1E,F)”.

Line 122: 25.1% is not “nearly half”, and 11/87 does not correspond to the number of ORFans, which is stated to be “eight”. I am confused.

Answer: sorry for this mistake. The number of unknown proteins and/or ORFans were corrected:

“Proteomic analysis of Tupanvirus soda lake particles revealed 127 proteins, nearly half (67/127 = 52.8%) of which are unknown and eight of which are encoded by ORFans (11/127 = 8.6%) (SI 3).”

Line 123 SI 3: This Table SI 3 is better if it describes gene function annotations. In addition, I do not understand the meaning of blank separators in the table.

Answer: at column B (ID/size (aa)) we show the identification of the ORFs; and at column E (description) we describe the predicted function of each ORF. In

the middle of blank separations there is information about the gene category, because we decide to separate the proteins/genes considering their function: DNA topology and repair, oxidative pathways, particle structure, protein/lipid modification, transcription and others.

Line 123: The sentence "Although 62 Tupanvirus virion proteins were not found by proteomics, either in Mimivirus or in Cafeteria roenbergensis virus particles, there are no distinct clues about which protein(s) could be associated with the tupanvirus fibrils and tail structure." needs rewording because it is impossible that Tupanvirus virion proteins are found in Mimivirus or in CroV particles. A word such as "homologs" may be missing here.

Answer: we agreed. The sentence was modified accordingly.

*"Although proteins homologous to 62 **Tupanvirus virion proteins homologous** were not found by proteomics, either in Mimivirus or in Cafeteria roenbergensis virus particles, there are no distinct clues about which protein(s) could be associated with the tupanvirus fibrils and tail structure."*

Reviewer #5 (Remarks to the Author):

This manuscript reports the discovery and characterization of two strains of tupanvirus, a giant dsDNA virus that is related to mimiviruses but presents several previously unseen features. The most striking difference to mimiviruses is the presence of a long, fiber-coated tail that appears to be hollow. Tupanvirus is also the largest mimivirus discovered so far, with a ~1.5 Mb dsDNA genome and 775 genes unique not found in other mimiviruses. The translation gene set of these viruses is impressive and suggests that tupanviruses are highly efficient in taking over host cell protein biosynthesis. The authors also report a cytotoxicity of the viral particles, even in non-permissive host cells such as Tetrahymena.

This study is highly fascinating, reports novel findings, and is written in a comprehensible manner, although language editing is recommended. As described below, the discussion could be modified to include alternative evolutionary scenarios.

Answer: Dear Reviewer, thank you very much for your comments and suggestions.

- Abstract, page 2. "Translation is the canonical frontier between the cell world and the virosphere" – Would the authors not consider energy production an equally important frontier?

Answer: This is an interesting topic. Some cellular organisms belonging to obligatory intracellular parasites lifestyle lack many genes related to energy production, and this would be one of the main causes of their parasitic lifestyle (Eg. *Encephalitozoon cuniculi*). Nevertheless we adjusted this sentence as suggested.

"Translation is one of the canonical frontiers between the cell world and the virosphere."

- Page 4. The authors compare the viral factory to a nucleus. Can this statement be specified in more detail, i.e. what makes the viral factory nucleus-like?

Answer: as suggested, we clarified this statement:

"This nucleus-like viral factory has also recently been reported in bacteria and fuels the concept of a virocell^{14,15}. In that perspective, viral factories actively producing the progeny could be considered as the nuclei of virocells^{14,15}."

- Page 5. This reviewer doubts that readers are familiar with the term "rhizome", and even though a reference is given, the term should be explained.

Answer: as suggested, we defined this term. In addition, a good reference about this term is cited in this sentence (number 17):

"The rhizome¹⁷ of tupanvirus (graphical representation of gene-by-gene best hits) revealed sequences from mimiviruses of amoebae (~42%) and klosneuviruses (~8%) as their main best hits."

An example of rhizome analysis application is given in SI10

- Conclusions, page 11. Regarding the "more generalist lifestyle" and the "reductive evolution" of the mimivirus ancestor, why do the authors insist on such a lopsided interpretation? Is it not equally likely that the mimivirus ancestor was in the process of acquiring genes (including translation genes), which proved advantageous to its propagation, and that the tupanvirus lineage further extended its host range and continued to incorporate additional genes, whereas the mimiviruses described so far had a narrower host range, for which a not-so-extensive translational gene complement was sufficient?

Answer: At conclusions we just hypothesized, it was not a statement. Nevertheless, we added an alternative scenario to this history:

Considering that tupanviruses comprise a sister group to amoebal mimiviruses, we can hypothesize that the ancestors of these Mimiviridae clades had a more generalist lifestyle and were able to infect a wide variety of hosts. In this view, the ancestors of tupanviruses (and maybe of amoebal mimiviruses) might have already been giant viruses that underwent reductive evolution, although some genes could have been acquired over time, as previously suggested for other mimiviruses²⁷. A reductive evolution pattern is typical among obligatory intracellular parasites²⁸⁻³¹. In these cases, the organisms lose genes related to energy production, which is one of the main reasons for their obligatory parasitic lifestyle. In an alternative scenario, a simpler ancestor could have substantially acquired genes over time and became more resourceful, being

able to infect a broader host-range. Nevertheless, tupanvirus presents the most complete translational apparatus among viruses, and its discovery takes us one step forward in understanding the evolutionary history of giant viruses.

Also, would tupanvirus no longer be placed in the virosphere if it had ribosomes?

Answer: this point is quite philosophical and controversial, therefore we removed this statement.